# Patterns and driving forces of dimensionality-dependent charge density waves in 2H-type transition metal dichalcogenides

Dongjing Lin[1], Shichao Li[1], Jinsheng Wen[1,2], Helmuth Berger[3], László Forró[3], Huibin Zhou[4,5], Shuang Jia[4,5], Takashi Taniguchi[6], Kenji Watanabe[6], Xiaoxiang Xi[1,2 ✉] & Mohammad Saeed Bahramy[7,8 ✉]

Charge density wave (CDW) is a startling quantum phenomenon, distorting a metallic lattice into an insulating state with a periodically modulated charge distribution. Astonishingly, such modulations appear in various patterns even within the same family of materials. Moreover, this phenomenon features a puzzling diversity in its dimensional evolution. Here, we propose a general framework, unifying distinct trends of CDW ordering in an isoelectronic group of materials, $2H$-$MX_2$ ($M$ = Nb, Ta and $X$ = S, Se). We show that while $NbSe_2$ exhibits a strongly enhanced CDW order in two dimensions, $TaSe_2$ and $TaS_2$ behave oppositely, with CDW being absent in $NbS_2$ entirely. Such a disparity is demonstrated to arise from a competition of ionic charge transfer, electron-phonon coupling, and electron correlation. Despite its simplicity, our approach can, in principle, explain dimensional dependence of CDW in any material, thereby shedding new light on this intriguing quantum phenomenon and its underlying mechanisms.

[1] National Laboratory of Solid State Microstructures and Department of Physics, Nanjing University, Nanjing 210093, China. [2] Collaborative Innovation Center of Advanced Microstructures, Nanjing University, Nanjing 210093, China. [3] Institute of Condensed Matter Physics, École Polytechnique Fédérale de Lausanne, 1015 Lausanne, Switzerland. [4] International Center for Quantum Materials, School of Physics, Peking University, Beijing 100871, China. [5] Collaborative Innovation Center of Quantum Matter, Beijing 100871, China. [6] National Institute for Materials Science, 1-1 Namiki, Tsukuba 305-0044, Japan. [7] Quantum-Phase Electronics Center (QPEC) and Department of Applied Physics, The University of Tokyo, Tokyo 113-8656, Japan. [8] RIKEN Center for Emergent Matter Science (CEMS), Wako 351-0198, Japan. ✉email: xxi@nju.edu.cn; bahramy@ap.t.u-tokyo.ac.jp

Two-dimensional (2D) materials have become a fertile playground for the exploration and manipulation of novel collective electronic states. Recent experiments have unveiled a variety of robust 2D orders in highly crystalline materials ranging from magnetism[1,2] to ferroelectricity[3–5] and from superconductivity[6] to charge density wave (CDW) instability[7,8]. The appearance of the latter in the 2D limit has, in particular, attracted a great deal of attention. By definition, CDW is the ground state of a one-dimensional metallic chain, leading to a simultaneous periodic modulation of its charge density and lattice dimerization[9]. In the 2D limit and beyond, CDW does not follow a unique pattern even within the same family of materials with identical electronic properties. While it is believed that phonon vibrations at low temperatures play a major role in the emergence of CDW ordering, its underlying mechanism has remained an unsettled question for decades[10,11] and its relation with superconductivity is still a subject of debates[12,13]. The advent of layered 2D materials[14] ushers in a new era of exploring CDW in confined dimensions[15,16]. Mechanical exfoliation[8,17–19], molecular-beam epitaxy (MBE)[7,20–22], and chemical vapour deposition[23,24] have produced a plethora of 2D materials exhibiting CDWs. Dimensionality reduction has been shown to enhance the CDW order in some of them[8,20,22,24] while suppress it in others[7,18,19,23,25]. Even for the same compound[7,8,20,21], results from different studies are scattered. For better understanding and control of the CDW states, it is thus imperative to determine the key factors governing the intrinsic effects of dimensionality on CDWs.

The group-V transition metal dichalcogenides (TMDs) $2H\text{-}MX_2$ ($M = $ Nb, Ta and $X = $ S, Se) are among the most studied CDW systems[10,11]. They all share similar crystal structures at high temperatures with isoelectronic chemical properties. At low temperatures, they behave differently, manifesting as CDW patterns unique to each compound. More interestingly, $2H\text{-}NbS_2$ stands out as CDWs are known to be absent in its bulk limit[26]. In a broad picture, two constraints facilitate the emergence of CDW instability and its diversity in this group of materials. The first is the fact that TMDs, in general, possess a quasi-2D electronic structure regardless of their thickness. This is due to the weak van der Waals stacking of $MX_2$ layers, allowing resonance of phonon vibrations along certain crystalline directions. Second, the metallic TMDs, as will be discussed later, exhibit a strong tendency to form additional covalent bonding between their chalcogens to compensate for the deficient charge transfer from the existing transition metals. When combined, these two can modulate the otherwise uniform charge distribution of metallic bands, such that the distorted state is chemically more robust.

In this work, we report systematic Raman scattering spectroscopy of the $2H\text{-}MX_2$ ($M = $ Nb, Ta and $X = $ S, Se) family. We show disparate dimensional dependence of the CDW properties in these materials, when the samples are reduced from their bulk limit to atomically thin layers. Based on these experimental observations, we establish a general framework that unifies all possible driving forces behind CDW ordering in 2D materials. We show that in principle, three major factors orthogonally govern the fate of CDW: ionic charge transfer, electron–phonon coupling, and the spreading extension of the electronic wave functions. We further explain how these factors compete with each other and manifest in CDW as such systems approach their bulk limit. The simplicity of our approach, combined with its success in accounting for such sophisticated features, paves a new way to study CDW and exotic quantum phenomena emerging from it in low dimensions.

## Results

### Chemical bonding in $2H\text{-}MX_2$.
The prototypes chosen here all crystalize in the so-called $2Ha$ structure[10] (Fig. 1a). The transition metal and chalcogen atoms form an $X$–$M$–$X$ sandwich in the trigonal prismatic coordination, henceforth called a monolayer. The unit cell for the bulk crystals consists of two such monolayers rotated 180° from each other and stacked with the transition metal atoms aligned along the $c$-axis. Common to all these compounds is a mismatch of valency between the central $M$ ion (3+) and the neighbouring chalcogens (each in 2− state). As such, they all show a metallic behaviour with a mixed ionic-covalent character. Figure 1a illustrates three major pathways for charge distribution in such a metallic environment: (1) an incomplete ionic charge transfer between the transition metal cation and its adjacent chalcogen anions within a monolayer (denoted as $\Delta Q_I$ and determined by their electronegativity difference, see Supplementary Fig. 1), forming the ionic $M$–$X$ bonds; (2) intralayer axial and planar $\sigma$-electron hoppings between the chalcogen $p$ orbitals, both forming covalent $X$–$X$ bonds; (3) interlayer $\sigma$-hoppings across the van der Waals gap.

The contribution of each channel to CDWs can be assessed according to whether the formed chemical bonds make the lattice harder or softer. In principle, a harder lattice is less susceptible to distortions, hence less amenable to CDW formation. This argument suggests the reduced tendency of CDW transitions in materials with sizable ionic charge transfer, as stronger ionic bonds make the lattice more rigid. However, the spatial extent of the ionic charge transfer crucially influences the strength of the ionic bonds. For example, the electronic bandwidth of the topmost valence band in monolayer $TaS_2$ is nearly 140% of that for monolayer $NbSe_2$ (Fig. 1b, c; for a full comparison also see Supplementary Fig. 2), thereby enabling a more extended charge distribution (as compared in Fig. 1d, e). This is ascribed to the large spin–orbit coupling of heavy tantalum atoms as well as the increased screening of the Coulomb potential from their nuclei due to their additional shells of semi-core and core states. Because of the extended charge distribution, despite the substantial total ionic charge transfer for $TaS_2$ shown in Fig. 1f, multiple bonds are involved in this sharing. Hence, its net effect on the lattice is counteracted in favour of CDW transition. In contrast, intralayer covalency, more pronounced in Nb-based compounds, is unfavourable for CDWs, as it pins the layers to each other, and thus, hardens the whole lattice. Interlayer $\sigma$-hopping works in unison with intralayer covalent bonding to account for the dimensionality-dependent CDWs in these compounds. This complexity in charge distribution obviously hinders the predictability of CDW phase transition in general and its dependence on dimensionality, specifically. However, as we discuss later, behind such chaos there appears to be an order that can be rationalized purely based on the chemistry of metals. Before that, let us first present the characteristic CDW features that we have observed experimentally in these compounds.

### Electrical transport characterization of bulk samples.
High-quality single crystals were chosen for our experiments. Their temperature-dependent resistance data in Fig. 2a show typical metallic behaviour, in stark contrast to the metal–insulator transition expected for Peierls instability. CDW transitions manifest as a weak hump at ~30 K for $NbSe_2$ and as clear kinks at ~80 and ~120 K for $TaS_2$ and $TaSe_2$, respectively. $NbS_2$ lacks CDW order and thus does not show an anomaly in the resistance. These temperatures for the incommensurate CDW transition ($T_{CDW}$) as well as the superconducting transitions at a lower temperature are consistent with those established in the literature[27]. Atomically thin samples were mechanically exfoliated from the bulk crystals, with special care taken to minimize sample degradation (see Methods).

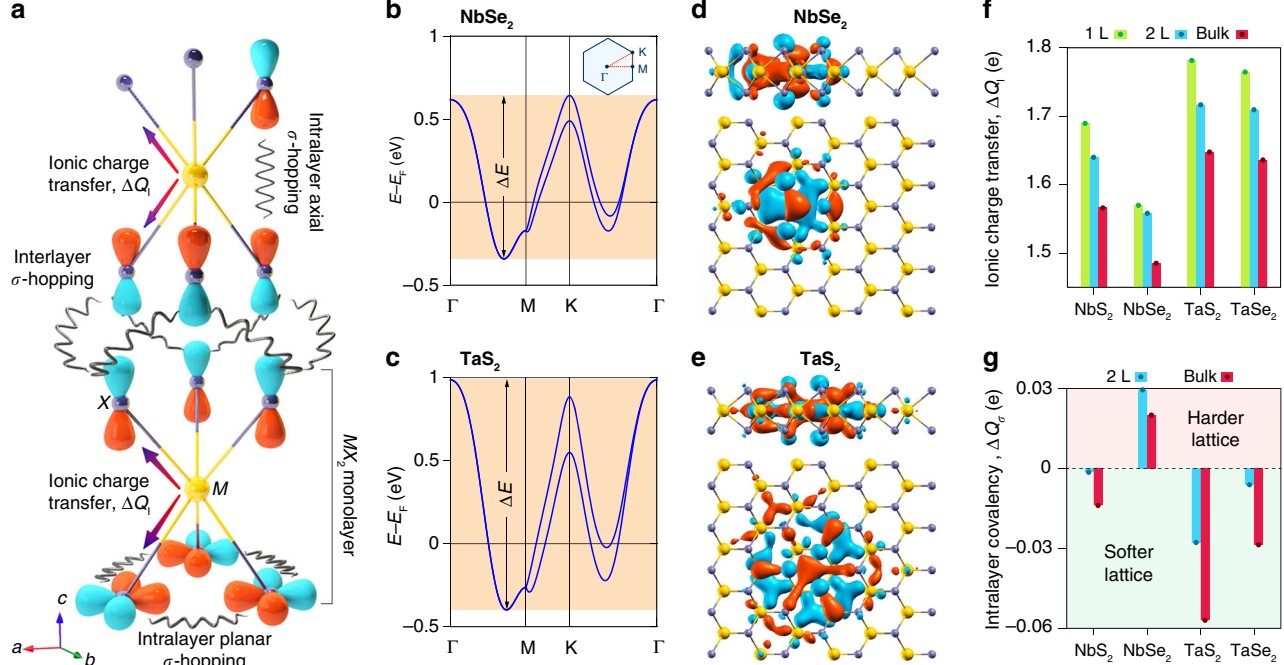

**Fig. 1 Chemical-bonding mechanism in 2H-MX₂. a** Schematic illustration of the charge transfer channels, including intralayer ionic charge transfer $\Delta Q_l$, intralayer covalent bonding (axial $\sigma$-hopping and planar $\sigma$-hopping), and interlayer $\sigma$-hopping. **b, c** Electronic band structure of monolayer NbSe₂ and TaS₂ obtained from first-principles calculations. $\Delta E$ and the corresponding shaded area denote the bandwidth of the topmost valence band. The inset in panel **b** represents the Brillouin zone and its high syemmetry $k$-points as well as the path (red dotted line) along which the electronic dispersions are calculated. **d, e** Calculated real-space spreads of the Wannier orbitals corresponding to the topmost valence bands in monolayer NbSe₂ and TaS₂, as viewed along the in-plane and out-of-plane directions. Orange and blue represent positive and negative regions of the electronic wave functions, respectively. **f, g** Layer number dependence of the ionic charge transfer $\Delta Q_l$ and the change of the intralayer covalency $\Delta Q_\sigma$ upon increasing the layer number from a monolayer for all compounds.

**Raman characterization of bulk and atomically thin samples.**
We used Raman scattering to obtain both CDW signatures and phonon information[28]. Figure 2b shows room temperature Raman spectra of the bulk compounds, collected in the collinear (XX) and cross (XY) polarization configurations. In the back-scattering geometry of our experiment, the former detects both $A_{1g}$ and $E_{2g}$ symmetry while the latter only couples to $E_{2g}$ (ref. [8]). Four common features are noted in the data: (1) a rigid layer mode with $E_{2g}$ symmetry (labelled as shear mode), which corresponds to interlayer shearing vibration and distinguishes the 2H from 1T and 3R polytypes; (2) an $A_{1g}$ phonon mode, which only involves the chalcogen atoms vibrating against each other along the $c$-axis; (3) an $E_{2g}$ phonon mode, which involves the transition metal and chalcogen atoms vibrating along opposite directions within the layer plane; (4) a broad two-phonon scattering peak commonly assigned as due to the CDW soft phonons[28–30].

The vibration patterns for the $A_{1g}$ and $E_{2g}$ modes suggest that the ionic bonds dominate their eigenfrequencies. As shown in Fig. 2c, both the $A_{1g}$ and $E_{2g}$ modes are found to have much higher frequencies in S-based 2H-MX₂ compounds than in Se-based ones. Similar results are found for the monolayer samples (Fig. 3e, f). To confirm this, we show in Fig. 3a–d the calculated phonon dispersions for all four monolayer 1H-MX₂ ($M$ = Nb, Ta and $X$ = S, Se) compounds. In a glance, one can notice a systematic difference between the S- and Se-based compounds. In the former, the optical modes appear at much higher frequencies, and the gap between the $A_{1g}$ and $E_{2g}$ modes at $\Gamma$ point (i.e. **q** = (0, 0, 0)) is much larger. Such a hardening of phonon modes is clear evidence that the chemical bonding in S-based compounds is more ionic (and thus stronger) than that in Se-based compounds. To be more precise, the higher electronegativity of

sulfur enables it to gain more charge from the $M$ site, thereby making the resulting $M$–S bond harder than its $M$–Se counterpart in 1H-$M$Se₂. This is consistent with the trend of ionic charge transfer shown in Fig. 1f. We note that in these calculations, to avoid negative frequencies, we have deliberately considered a relatively large broadening factor ($\sigma = 0.03$ Ry) for treating the distribution of electrons at the Fermi level[31] (see Methods). Nevertheless, the calculated values for the $A_{1g}$ and $E_{2g}$ mode frequencies at $\Gamma$ point agree reasonably well with our experimental data (Fig. 3e, f), even confirming the fact that NbSe₂ is the only compound in this family in which the $A_{1g}$ mode lies below the $E_{2g}$ mode. The largest deviation is found for NbS₂ where the calculations underestimate the $E_{2g}$ mode by ~19%. We attribute this to the localized nature of wave functions in this material, requiring exchange-correlation approximations beyond standard density-functional theory (DFT) as well as more accurate pseudo-potentials.

Figure 2d compares the 300 and 4 K spectra, with new modes emerging at low temperature in all compounds except NbS₂. These are the amplitude modes or zone-folded modes unique to the CDW phase[8,29,30,32,33]. The former features characteristic soft-ening and broadening upon approaching $T_{CDW}$ from below, while the latter shows only a minute change of the frequency. Fig. 4j–l show the layer number dependence of these modes at 4 K for NbSe₂, TaSe₂, and TaS₂, respectively. We focus on the amplitude modes, which are the most pronounced features in the spectra below 100 cm⁻¹. NbSe₂ exhibits a single broad peak that shifts from below 50 cm⁻¹ in the bulk to 73 cm⁻¹ in the monolayer, while the intensity is less affected. In contrast, both TaSe₂ and TaS₂ exhibit multiple sharper amplitude modes below 100 cm⁻¹. Each peak shows a weak layer number dependence for its

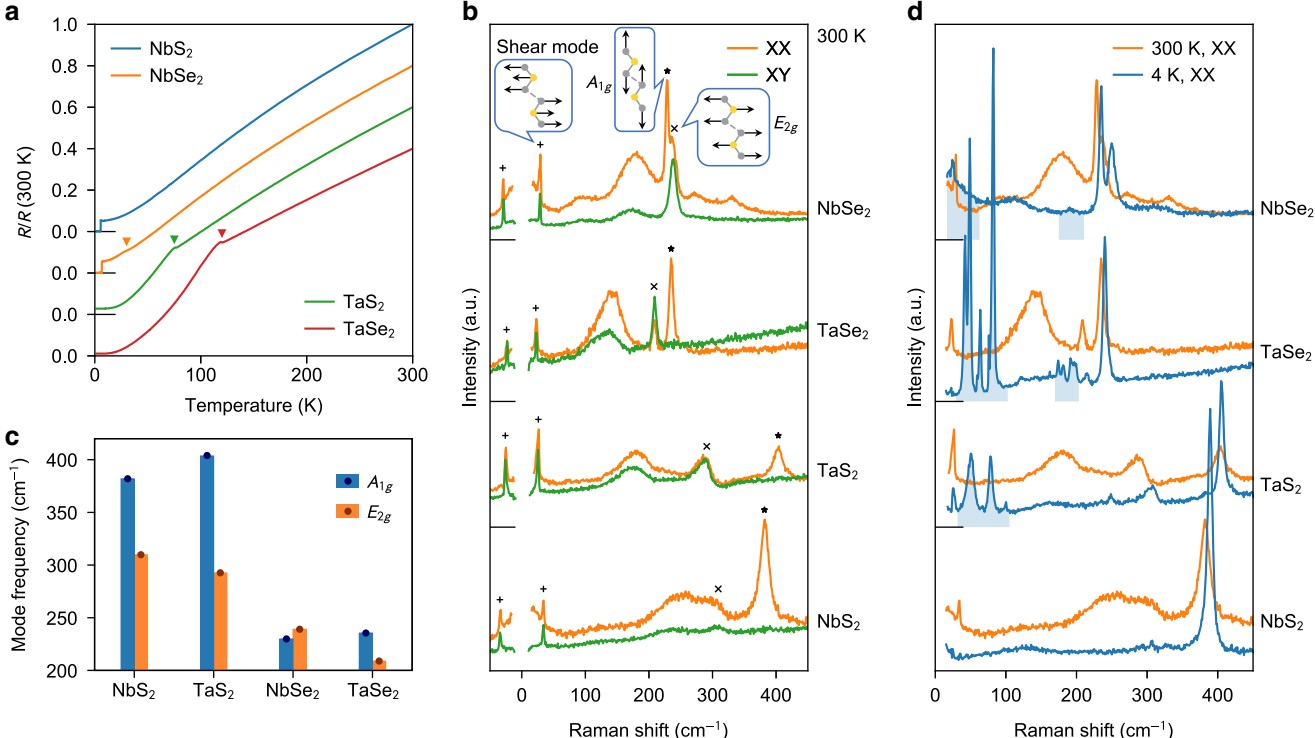

**Fig. 2 Characterizations of phonon modes in bulk 2H-MX₂. a** Temperature dependence of the electrical resistance for all compounds, normalized to their respective values at 300 K. The triangles mark the anomalies due to CDW transitions. **b** Raman spectra for all compounds collected in the collinear (XX) and cross (XY) polarization configurations at 300 K. The first-order phonon scattering peaks are indicated by daggers for the shear modes (Stokes and anti-Stokes), crosses for the $E_{2g}$ modes, and stars for the $A_{1g}$ modes. The corresponding displacement patterns are illustrated in the balloons. **c** Compound dependence of the $A_{1g}$ and $E_{2g}$ mode frequencies analysed by peak fitting of the data in **b**. **d** Comparison of the Raman spectra at 300 and 4 K for the collinear polarization configuration for all compounds. CDW-induced modes are highlighted by the shaded regions. Data in **a**, **b**, and **d** are shifted vertically for clarity, with the new origins located at the crossings of the left axes and the long horizontal bars.

frequency, while the intensity dramatically diminishes in atomic layers. The blueshift of the mode frequency (suppression of the mode intensity) correlates well with enhanced (reduced) $T_{CDW}$ in atomically thin NbSe₂ (TaSe₂ and TaS₂), as will be detailed below. The completely different behaviours of the amplitude modes in the Nb- and Ta-based compounds call for further investigations.

We do not observe any new modes at low temperature in NbS₂ down to the monolayer (Supplementary Fig. 3), therefore concluding the entire absence of CDW in this compound. Previous work has shown latent CDW in bulk NbS₂ due to the strong anharmonicity of the lattice potential[26]. Our temperature-dependent study of the $A_{1g}$ mode frequency shows that NbS₂ indeed exhibits the largest degree of lattice anharmonicity among the four compounds (Supplementary Note 4). Our calculation shown in Fig. 1f suggests enhanced ionic bonding upon reducing the layer number in NbS₂. CDWs, therefore, do not emerge in atomically thin samples, although a recent study predicts otherwise[34].

Temperature and layer-number-dependent Raman measurements show systematic evolution of the amplitude modes in NbSe₂, TaS₂, and TaSe₂ (Supplementary Fig. 3). To facilitate comparison of the mode intensity among different samples, we transform the raw spectra $I$ to $I/I_0 - 1$, where $I_0$ is a high-temperature spectrum far above $T_{CDW}$. The normalized temperature-dependent Raman scattering intensity maps for samples of various thickness are shown in Fig. 4a–i. The amplitude modes are accentuated in red, which diminish into the background upon increasing temperature. For NbSe₂, we clearly identify a strong enhancement of $T_{CDW}$ in the bilayer and monolayer, consistent with a previous Raman study using a

different excitation laser energy[8]. For TaSe₂ and TaS₂, although the mode intensity is significantly weakened in the atomically thin samples, the transition temperature appears similar to that of the bulk.

Figure 5a summarizes the layer number dependence of $T_{CDW}$ for each compound, estimated from the temperature dependence of the amplitude mode intensity, integrated with respect to a featureless background (Supplementary Note 6). The figure clearly shows enhanced (suppressed) $T_{CDW}$ in NbSe₂ (TaSe₂ and TaS₂) when approaching the monolayer limit. The trends are further confirmed by analysing the amplitude mode frequency and the zone-folded mode intensity (Supplementary Notes 7–9). NbS₂ does not show Raman signature of CDWs for the measured bulk, bilayer, and monolayer samples, hence their vanishing $T_{CDW}$.

**Unified theory for CDWs in 2H-MX₂.** The disparate thickness-dependent $T_{CDW}$ observed for the isostructural and isoelectronic compounds in the same material family is highly unusual. Since the significant two-phonon scattering peak in these compounds originates from the longitudinal acoustic phonon branch exhibiting Kohn anomaly[28–30], and because its frequency and temperature variation show rather weak thickness dependence (Supplementary Note 10), we infer a nearly thickness-independent CDW wave vector for NbSe₂, TaSe₂, and TaS₂. The thickness dependence of $T_{CDW}$ is, therefore, not related to different forms of superlattice. Instead, it appears to originate from the intrinsic chemical properties of the ingredients, constituting these materials. First of all, the existence of CDWs in monolayer NbSe₂, TaSe₂, and TaS₂ is permitted by their incomplete ionic bonding. As we explained earlier, this is primarily due

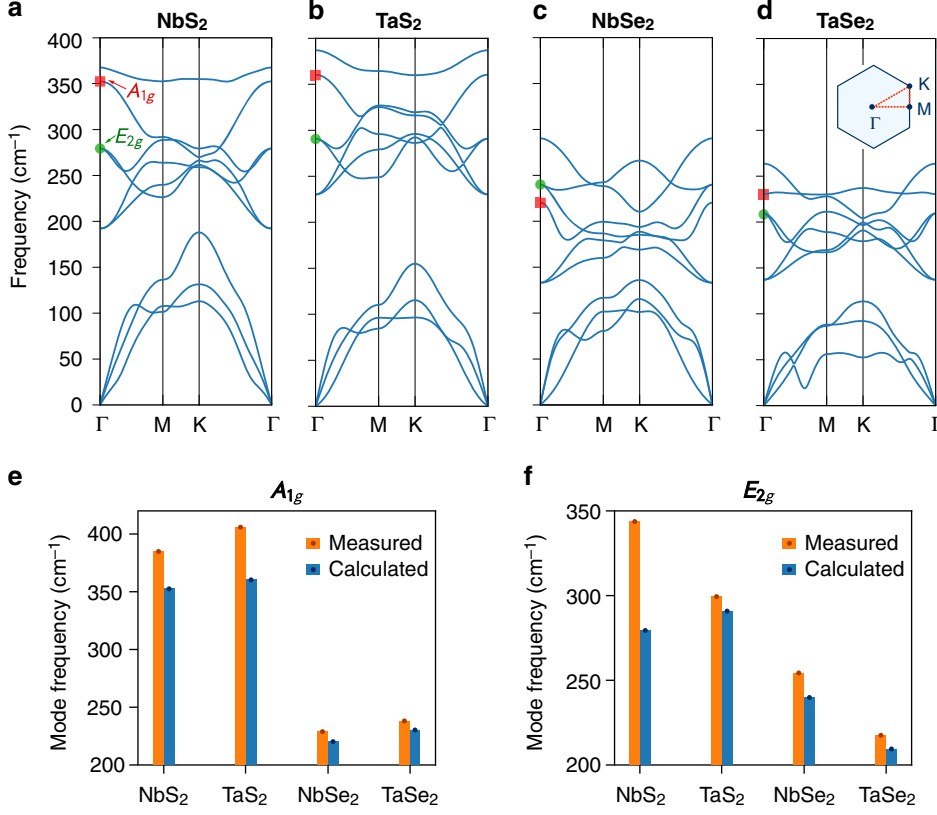

**Fig. 3 Phonon dispersions and Raman-active phonon modes in monolayer 1H-MX₂. a–d** Calculated phonon dispersions for the monolayer 1H-MX₂ compounds. The red squares and green circles indicate the expected locations of the Raman-active phonon modes $A_{1g}$ and $E_{2g}$, respectively. The inset in panel **d** represents the Brillouin zone and its high syemmetry $q$-points as well as the path (red dotted line) along which the phonon dispersions are calculated. **e** Comparison of the measured $A_{1g}$ mode frequency analysed from the data at 4 K in Supplementary Fig. 3 and the calculated mode frequency. **f** The corresponding results for the $E_{2g}$ mode.

to a mismatch between the valence state of $M$ cations and $X$ anions. Furthermore, the occurring ionic charge transfer is either small and spatially highly localized as in the case of NbSe₂ or significant but spatially extended as in the case of TaSe₂ and TaS₂. In going from a monolayer to bulk, the nature of bonding between the layers and their effect on the spread of the wave functions defines the fate of the CDW in each compound. If the interlayer bonding enhances the rigidity of each layer, $T_{CDW}$ is expected to show declining behaviour, as the thickness grows. On the other hand, if it weakens the initial bonding, the result will manifest as an enhancement of CDW instability.

The parameter which quantifies either of these trends is the intralayer covalency, $\Delta Q_\sigma$. We define this parameter as the differential charge at the chalcogen sites upon migration from a monolayer to a multilayer system. A positive $\Delta Q_\sigma$ means that the interlayer hopping has helped each chalcogen to gain more charge, thereby making the whole entity a harder lattice. Needless to say, a negative $\Delta Q_\sigma$ means the added layers have led to a softer lattice. Using atomic-orbital-like Wannier functions (see Methods), we have calculated $\Delta Q_\sigma$ for all four compounds in the bilayer and bulk configurations (Fig. 1g). As can be seen, both bilayer and bulk NbSe₂ show a significant gain in $\Delta Q_\sigma$, implying that their crystal structure is relatively harder than that of monolayer NbSe₂, and thus, less prone to CDW instability. We attribute this to the localized nature of wave functions in NbSe₂, mainly made up by the axial orbitals Se-$p_z$ and Nb-$d_{z^2}$ (see Fig. 1d). In a bilayer or bulk NbSe₂, this accordingly allows a direct $\sigma$-type interlayer hopping between these orbitals, making $\Delta Q_\sigma$ positive. More importantly, this additional bonding acts against the $A_{1g}$ phonon mode, resulting in a decrease in the

strength of electron–phonon coupling constant $\lambda$ as depicted in Supplementary Fig. 14. It then becomes clear why $T_{CDW}$ in NbSe₂ shows a negative thickness dependence.

TaS₂ and TaSe₂ behave differently. As shown in Fig. 1g, both compounds have a negative $\Delta Q_\sigma$ with lower values found for their bulk phases. Given their extended wave functions, the only channel allowing the interlayer coupling is $\sigma$-hopping between the chalcogen $p_z$ orbitals of neighbouring layers. As such, the initial charge residing on the chalcogen sites is partly shifted between the layers. This, in turn, leads to a softening of the original metal–chalcogen bonding and correspondingly an enhancement of CDW instability, as observed experimentally. Interestingly, the calculated $\Delta Q_\sigma$ for TaS₂ turns out to be more sensitive to the number of layers as compared with that obtained for TaSe₂. Logically, one then expects a higher $T_{CDW}$ in, for example, bulk TaS₂ than in bulk TaSe₂. In reality, however, TaSe₂ exhibits a higher $T_{CDW}$ than TaS₂ regardless of the number of the layers. To address this apparent inconsistency, we need to include another critical parameter in our consideration. That parameter is electron–phonon coupling $\lambda$. A large $\lambda$ can mediate CDW more effectively than a small $\lambda$. However, this requires a proper coupling between the phonon mode eigenvalues and electronic eigenvalues. If the phonon modes, responsible for CDW instability, have relatively high frequencies, such a coupling is not well established and, hence, a relatively small $\lambda$ is achieved. This is what occurs in TaS₂. As discussed earlier, for both $A_{1g}$ and $E_{2g}$ modes, the corresponding phonon frequencies are higher in TaS₂ than in TaSe₂ (Figs. 2c and 3e, f), suggesting that the former should have a smaller $\lambda$ than the latter. Our first-principles calculations for monolayer 1H-MX₂ confirm this indeed. In fact,

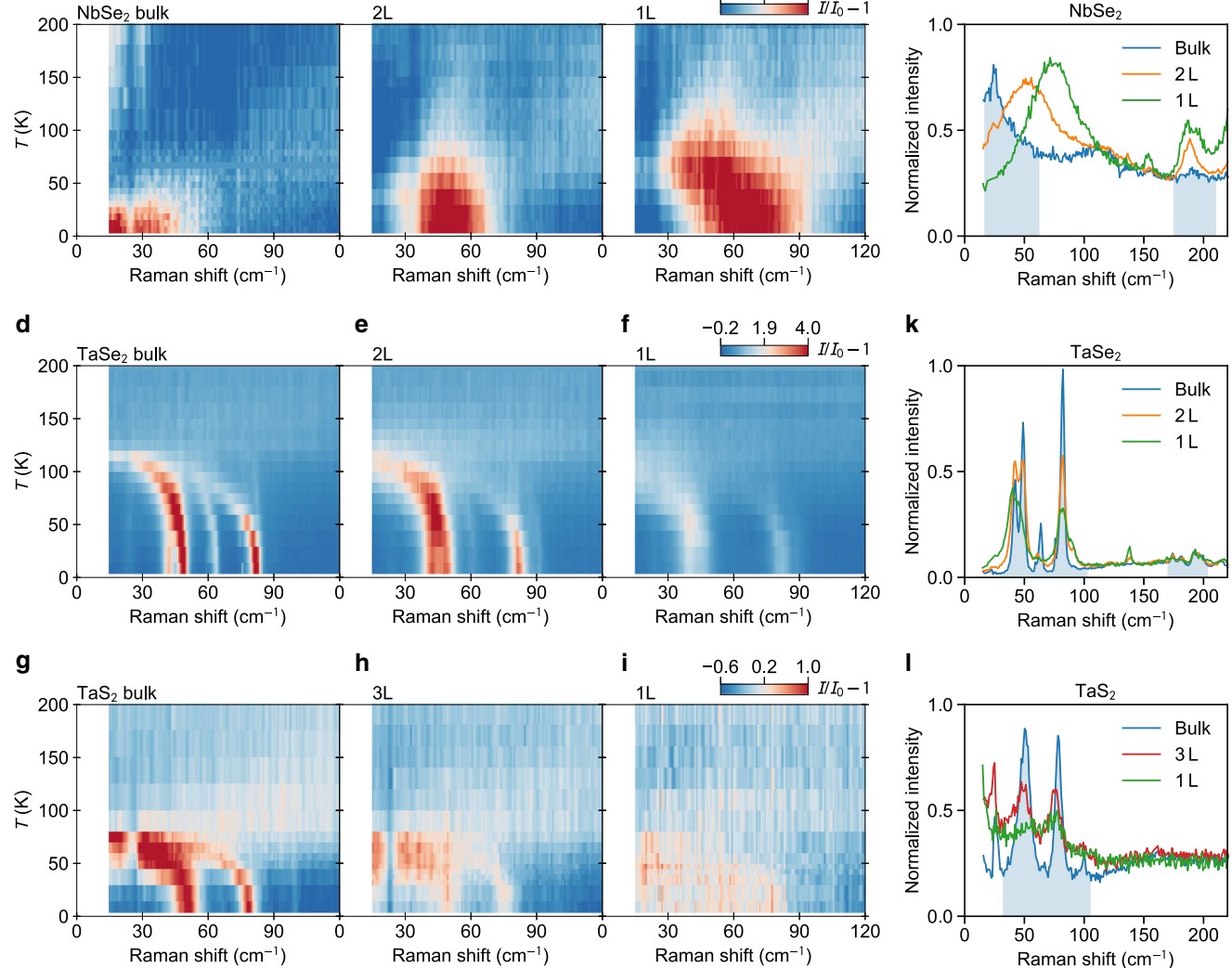

**Fig. 4 Raman signature of CDW transitions in NbSe₂, TaSe₂, and TaS₂. a–i** Temperature-dependent Raman scattering intensity maps for the three compounds with different thickness, all collected in the collinear polarization configuration. To compare different samples, each set of raw data $I$ are normalized to a corresponding high-temperature spectrum $I_0$ far above $T_{CDW}$, and unity is subtracted from the ratio to yield $I/I_0 - 1$. Data for the same compound share the same colour scale. **j–l** Comparison of the Raman scattering spectra at 4 K for samples of different thickness. In each panel, the spectra are normalized to match their background intensity between 150 and 220 cm$^{-1}$. CDW-induced modes in the bulk samples are highlighted by the shaded regions.

as shown in Fig. 5b, we find that $\lambda$ is always smaller in an S-based compound when compared to its Se-based counterpart. These calculated results agree well with the available $\lambda$ values reported for NbSe₂ (ref. [31]), TaS₂, and TaSe₂ (ref. [35]).

In a broader context, we can again attribute this tendency to the ionic charge transfer $\Delta Q_I$ we discussed earlier, implying a trade-off between $\Delta Q_I$ and $\lambda$. In addition to these two parameters, the spatial extension of the wave functions (represented here as $1/\langle r^2 \rangle$) plays an important role. The more localized are the wave functions, the less effective become both $\Delta Q_I$ and $\lambda$. This is because it enhances the electron–electron correlation, demobilizing the carriers and so preventing the formation of any superlattice charge ordering. As a result, if $\lambda$ is not large enough, CDW remains forbidden. This then solves the last piece of the puzzle, i.e. why NbS₂ is so robust against CDW instability. We note that van Loon et al.[36] have already given a detailed account for the competing nature of electron–phonon coupling versus the short- and long-range electron Coulomb interaction, and its prominent role in the absence of CDW instability in NbS₂.

Interestingly, they suggest that the interplay between these parameters strongly enhances both charge and spin susceptibilities in NbS₂, meaning that it is at the verge of collapsing to a CDW or even a spin density wave phase, if sufficient perturbations are introduced (for example, via local magnetic impurities)[36]. As such, NbS₂ appears to be an ideal candidate for exploring thermal and quantum fluctuations.

Altogether, the three parameters (1) ionic charge transfer, (2) electron–phonon coupling, and (3) the spatial extension of the electronic wave functions are the key components defining the fate of CDW ordering and its thickness dependence in this and potentially other layered materials. Thus, they can be used to create a unified phase diagram describing such instabilities in low-dimensional limits. We have schematically illustrated such a phase diagram in Fig. 5c. As can be seen, the interplay between these three parameters is expected to form an abyss-like shape, deepening at regions with low $\Delta Q_I$ and $\lambda$ and high $1/\langle r^2 \rangle$. Above the surface of the resulting terrain is a parameter-space within which the CDW is forbidden and below it is where CDW is

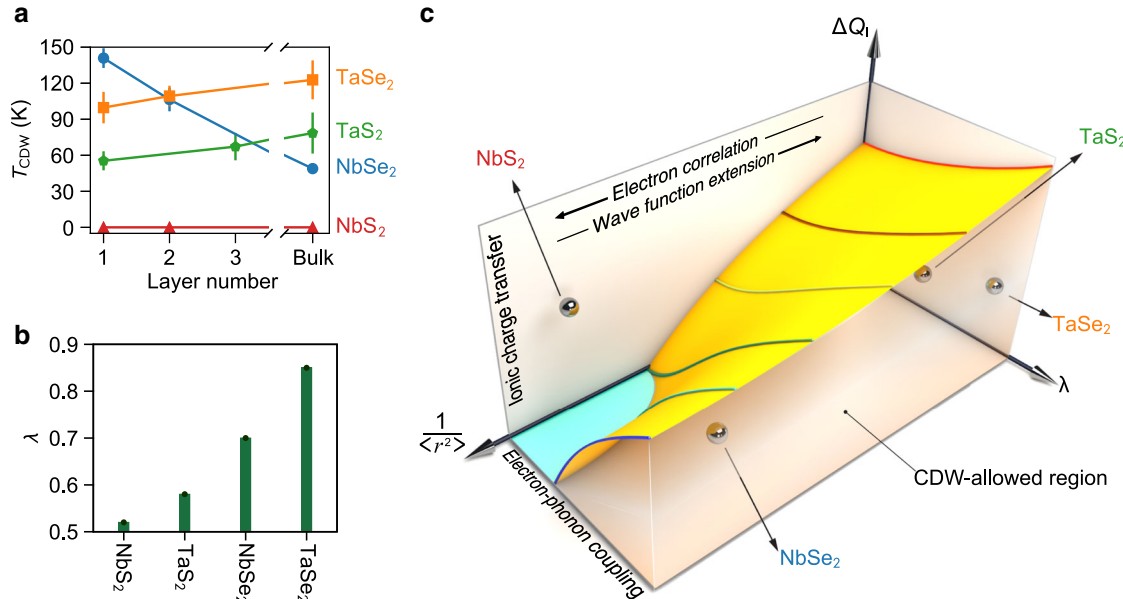

**Fig. 5 Thickness dependence of $T_{CDW}$ in 2H-MX$_2$. a** Layer number dependence of the CDW transition temperature $T_{CDW}$ for all compounds. Error bars are standard deviations obtained from the least-squares fits to the temperature-dependent amplitude mode intensity (Supplementary Fig. 9). **b** The calculated values of electron–phonon coupling constant $\lambda$ for monolayer 1H-MX$_2$. **c** Schematic illustration of the possible phase diagram describing the CDW response in a layered material in terms of ionic charge transfer $\Delta Q_I$, electron–phonon coupling constant $\lambda$, and the spatial extension of electronic wave functions $\frac{1}{\langle r^2 \rangle}$.

allowed to emerge. We also have shown the schematic location of each of the 2H-MX$_2$ compounds, studied here. Obviously, NbS$_2$ is the only member of this family appearing in the region where CDW is prohibited.

## Discussion

Lastly, we address some controversies in the studies of 2D CDWs in NbSe$_2$, TaSe$_2$, and TaS$_2$. For NbSe$_2$, a strongly enhanced $T_{CDW}$ from ~33 K in the bulk to ~145 K in the monolayer was previously reported in mechanically exfoliated samples on sapphire substrates[8] and confirmed here. This is in stark contrast to the almost unchanged $T_{CDW}$ in MBE-grown monolayer NbSe$_2$ on bilayer graphene[7]. For TaSe$_2$, we found a rather weak suppression of $T_{CDW}$ in the exfoliated monolayer, while MBE-grown monolayer TaSe$_2$ on bilayer graphene shows slightly enhanced $T_{CDW}$ with respect to its bulk value[22]. The exfoliated samples transferred on sapphire substrates are expected to exhibit intrinsic CDW properties. The MBE-grown samples may be affected by charge transfer from the underlying graphene substrate. For monolayer NbSe$_2$, such charge transfer can increase the intralayer covalency and suppress the strongly enhanced $T_{CDW}$. For monolayer TaSe$_2$, due to its extended charge distribution, charge transfer tends to accumulate between TaSe$_2$ and graphene, so that the intralayer covalency is reduced; CDW is therefore enhanced.

Interestingly, CDWs were found to be absent in epitaxial monolayer TaS$_2$ on Au(111) substrate[37], but persist in MBE-grown monolayer TaS$_2$ on graphene/Ir(111)[38] as well as in the exfoliated monolayer studied here. These findings provide further evidence that the CDW formation in such atomically thin TMDs is, in general, highly susceptible to the surrounding environment, which could be a neighboring layer or even a substrate. As we discussed earlier, the deficient charge transfer in metallic TMDs makes them active in finding new pathways for electron hopping so that the chalcogen $p$ orbitals can gain all electrons they need to form a closed shell system. Such pathways could, in principle, extend to a substrate. As an example, with Au(111) surface as a substrate, one can expect electron charge transfer from spatially extended 4$s$ electrons of Au to the chalcogens above them.

Meanwhile, the trigonal symmetry of Au(111) substrate enforces the original symmetry of pristine monolayer TaS$_2$, thereby avoiding any lattice deformation. Of course, hybridization between monolayer TaS$_2$ and the Au(111) substrate could significantly affect its electronic band structure[38–40], and correspondingly, its intrinsic properties. Alternatively, when the substrate is made of chemically inactive orbitals (such as $p_\pi$ states in graphene), the CDW is still inevitable.

In conclusion, our chemical-bonding framework provides an intuitive guide for boosting $T_{CDW}$, which is essential for developing CDW-based devices for applications. It may also be applied to explain the distinct dimensionality effects in the growing family 2D CDW materials[41]. Future work should address the connection of the current framework with the existing theories for CDWs, for instance, Fermi-surface nesting and momentum-dependent electron–phonon coupling[11,42].

## Methods

**Sample preparation**. Bulk single crystals were synthesized by the chemical vapour transport method. Atomically thin flakes were mechanically exfoliated from the bulk crystals on silicone elastomer polydimethylsiloxane stamps and transferred on sapphire substrates for Raman study. To minimize sample degradation, we prepared them in a glove box filled with nitrogen gas, followed by encapsulation with thin h-BN. The flake thickness was determined by the shear mode frequency in the Raman data (Supplementary Note 3).

**Characterizations**. Temperature-dependent Raman scattering measurements were performed using a home-built confocal optical setup, consisting mainly of a Montana Instruments Cryostation and a Princeton Instruments grating spectrograph equipped with a liquid-nitrogen-cooled charge-coupled device. Beam from a diode-pumped 532 nm laser was focused on the sample using a ×40 microscope objective. The backscattered light was collected using the same objective followed by a couple of Bragg notch filters, achieving a minimum cut-off of 15 cm$^{-1}$ in the collinear polarization configuration. The sample chamber was evacuated to high vacuum better than 10$^{-4}$ Pa throughout the experiment. Rapid temperature control was achieved using an Agile Temperature Sample Mount. To minimize laser heating, the incident power was kept below 0.1 mW for bulk samples and below 0.3 mW for thin flakes, and anti-Stokes lines were confirmed to be absent at the base temperature. Four-probe resistance measurements on the bulk crystals were conducted in an Oxford Instruments TeslatronPT system using the standard lock-in method.

**Calculations.** The electronic structure of $1H$-$MX_2$ monolayers was calculated within DFT using Perdew–Burke–Ernzerhof exchange-correlation functional[43] as implemented in Quantum Espresso program package[44–46]. We used the norm-conserving pseudo-potentials[47] and plane-wave basis set with cut-off energy of 75 Ry. The relativistic effects, including spin–orbit coupling, were fully considered. The Brillouin zone (BZ) was sampled by a $24 \times 24 \times 1$ $k$-mesh. The ionic charge transfer and hopping parameters were obtained by constructing a 22-band tight-binding model by downfolding the DFT Hamiltonian using maximally localized Wannier functions[48,49]. The $M$-$d$ and $X$-$p$ atomic orbitals were taken as projection centres.

For the calculation of phonon modes and the corresponding electron–phonon coupling parameters, we first fully optimized both the lattice parameters and atomic positions until the magnitude of the force on each ionic site was less than $10^{-5}$ Ry Bohr$^{-1}$ and the total energy was converged below $10^{-10}$ Ry. To avoid the negative phonon modes, resulting in an overestimation of $\lambda$, the Methfessel–Paxton smearing scheme with a relatively large broadening parameter $\sigma = 0.03$ Ry was used, as suggested in ref. [31]. The dynamical matrix was then calculated based on density-functional perturbation theory employing an $8 \times 8 \times 1$ $q$-mesh. Finally, we computed $\lambda$ using the interpolation scheme implemented in Quantum Espresso[46]. In this scheme[50], $\lambda$ is obtained by summing up the electron–phonon coupling arising from each individual phonon mode $\nu$ at all available phonon wave vectors $\mathbf{q}$, i.e. $\lambda = \sum_{\mathbf{q}\nu} \lambda_{\mathbf{q}\nu}$. $\lambda_{\mathbf{q}\nu}$ is expressed as follows:

$$\lambda_{\mathbf{q}\nu} = \frac{1}{N(\varepsilon_F)\omega_{\mathbf{q}\nu}} \sum_{nm} \int_{BZ} \frac{d\mathbf{k}}{\Omega_{BZ}} |g_{mn}^{\nu}(\mathbf{k}, \mathbf{q})|^2 \delta(\varepsilon_{n\mathbf{k}} - \varepsilon_F)\delta(\varepsilon_{m\mathbf{k}+\mathbf{q}} - \varepsilon_F),$$

where $\omega_{\mathbf{q}\nu}$ is the phonon frequency for mode $\nu$ at the wave vector $\mathbf{q}$; $\varepsilon_{n\mathbf{k}}$ is the eigenvalue of the $n$th Kohn–Sham eigenfunction $\psi_{n\mathbf{k}}$ at wave vector $\mathbf{k}$ in the BZ; $\varepsilon_F$ is the Fermi level; $N(\varepsilon_F)$ corresponds to the density of states per spin at $\varepsilon_F$; and $\Omega_{BZ}$ is the volume of the BZ. Above, $g_{mn}^{\nu}(\mathbf{k}, \mathbf{q})$ is the first-order electron–phonon matrix element defined as

$$g_{mn}^{\nu}(\mathbf{k}, \mathbf{q}) = \frac{1}{\sqrt{2\omega_{\mathbf{q}\nu}}} \langle \psi_{m\mathbf{k}+\mathbf{q}} | \partial_{\mathbf{q}\nu} V | \psi_{n\mathbf{k}} \rangle$$

with $\partial_{\mathbf{q}\nu} V$ being the first derivative of the self-consistent potential associated with phonon $\omega_{\mathbf{q}\nu}$.

## Data availability
The data in Fig. 5a are available in Supplementary Table 1. Other data that support the findings of this study are available from the corresponding authors upon request.

## Code availability
The calculations discussed in this study are performed using Quantum Espresso program package[44–46]. This program is open source and can be downloaded at http://www.quantum-espresso.org, freely. In the Source Data, we have provided detailed instruction on the procedures needed for the electronic and phonon calculations. Example input files for the reproduction of electronic structure, phonon dispersions, and electron–phonon coupling of NbS$_2$ are also presented. For additional information, please contact Mohammad Saeed Bahramy (bahramy@ap.t.u-tokyo.ac.jp).

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

## Acknowledgements

This work was supported by the National Key Research and Development Program of China (Grant Nos. 2018YFA0307000 and 2017YFA0303201), the National Natural Science Foundation of China (Grant Nos. 11774151, 11822405, and 11674157), the Natural Science Foundation of Jiangsu Province (Grant No. BK20180006), and Japan Science and Technology Agency (CREST, JST, Grant No. JPMJCR16F1). The work in Lausanne was supported by the Swiss National Science Foundation. The crystal growth in Peking University was supported by the National Natural Science Foundation of China (Grant Nos. U1832214 and 11774007) and the National Key Research and Development Program of China (2018YFA0305601). Growth of hexagonal boron nitride crystals was supported by the Elemental Strategy Initiative conducted by the MEXT, Japan and the CREST (JPMJCR15F3), JST.

## Author contributions

X.X. and M.S.B. conceived the project. D.L. and X.X. performed the experiments. S.L., J.W., H.B., L.F., H.Z., and S.J. provided the transition metal dichalcogenides crystals. T.T. and K.W. grew the h-BN crystals. D.L. and X.X. analysed the experimental data. M.S.B. performed the first-principles calculations. X.X. and M.S.B. interpreted the results and co-wrote the paper, with comments from all authors.

## Competing interests

The authors declare no competing interests.
