## [Peer Review File · Nature Communications]

Reviewers' comments:

Reviewer #1 (Remarks to the Author):

Dear editor,

I have read the manuscript "Patterns and driving forces of dimensionality-dependent charge density waves in 2H-type transition metal dichalcogenides" by Dongjing Lin et al. In the paper, they investigate the phonon modes of TaS₂, TaSe₂, NbS₂ and NbSe₂ using Raman spectroscopy. This characterization is carried out both at low and at room temperature and for monolayer, bilayer and bulk materials. These experimental results are supplemented by ab-initio and theoretical analysis. The main goal of the paper is to gain more understanding of the differences in) the CDWs observed in these materials. Throughout the paper, NbS₂ is the odd one out: it does not have a CDW at any temperature.

I found the manuscript a pleasure to read. The experimental results are laid out clearly and comprehensively, covering a relevant and diverse set of experimental conditions. Transition metal-dichalcogenides are an active topic of current experimental and theoretical research and this work is a very worthy contribution to the field, both in terms of I believe it should be published in Nature Communications. I do have some questions and remarks, which follow below.

* Around line 70: This paragraph contains a discussion of mechanisms contributing for or against CDW. However, the reason why this class of materials wants to form a CDW in the first place is left rather open here.

* Around line 205: Is this analysis based on the equation $\lambda = 2g^2 / w_{\text{phonon}}$? In that case, it might be useful to include this formula.

* More generally, including a clearer definition of the electron-phonon coupling λ in the main text could be helpful, also to put Fig 4b into context.

* The authors have performed ab-initio calculations, for the band structure and the electron-phonon coupling. I would be interested to know about the ab-initio phonon spectra and their relation with the experimentally observed Raman spectra.

* In the discussion of Ref 34, around line 255: Hall et al. (ACS Nano 2019, 13, 9, 10210-10220) discuss the role of the environment in TaS₂ monolayers in more detail. They find that it is not just charge transfer but also hybridization between monolayer and substrate that is responsible for the absence of CDWs on gold substrates.

* It would be useful to compare these Raman spectroscopy results with those available in the recent literature, e.g. Phys. Rev. Lett. 122, 127001(2019), Phys. Rev. B 100, 165414 (2019), Phys. Rev. B 98, 165109 (2018) and Phys. Rev. B 97, 094502 (2018)

Smaller comments:

* Fig 1 and page 3: "planner hopping" -> "planar hopping"?

* Fig 1 (g): I would write the label as "bulk has harder/softer lattice", so that the panel can be understood without the caption. In the current version, it is not clear which situation has the harder lattice.

* Fig 1f and similar figures: It does not really make sense to connect the data points by lines here, and I find it a bit distracting.

* Line 92, "Pierels"-> "Peierls"

Reviewer #2 (Remarks to the Author):

This work experimentally investigates the CDW transition in thin exfoliated sheets of NbSe₂, TaSe₂ and TaS₂, based upon the emergence of the corresponding amplitude modes in the Raman spectrum. A major finding is that T(CDW) increasing (decreases) with decreasing sheet thickness for NbSe₂ (TaSe₂ and TaS₂).

As there are indeed partially contradictory theoretical predictions and also experimental observations reported in the literature on CDWs in this (and similar) type(s) of material, the present study makes a novel and valuable contribution to this field. The detailed Raman experiments seem to be carried out carefully (it is especially important that the bulk crystals were exfoliated in a glove box and subsequently encapsulated by h-BN), and also the data evaluation looks reasonable.

Furthermore, the three parameter-based model (fig. 1a) proposed to account for the presence or absence (like in NbS₂) of CDW instability, and the thickness dependence of T(CDW) for the three different compounds, makes sense and is - despite its simplicity - in quite good agreement with the experimental observations. Although no fundamentally new theoretical result is provided, the presented model can nevertheless help to further clarify relevant issues on CDWs in 2D materials. Finally, also the mentioned experimental factors (in particular the role of the substrate) are indeed likely to have an influence on T(CDW) in these compounds.

Note: In accordance with Nature Communications format and style, we have rephrased the Abstract and Introduction of our manuscript. We have also added section and subsection titles to meet the editorial requirements. These changes do not affect our original discussions. We appreciate both reviewers for taking the time to evaluate our work, and hope with revisions, described below, we have been able to address all their valuable comments and suggestions.

RESPONSE TO REVIEWER 1

Dear editor,

I have read the manuscript “Patterns and driving forces of dimensionality-dependent charge density waves in 2H-type transition metal dichalcogenides” by Dongjing Lin et al. In the paper, they investigate the phonon modes of TaS₂, TaSe₂, NbS₂ and NbSe₂ using Raman spectroscopy. This characterization is carried out both at low and at room temperature and for monolayer, bilayer and bulk materials. These experimental results are supplemented by ab-initio and theoretical analysis. The main goal of the paper is to gain more understanding of the differences in the CDWs observed in these materials. Throughout the paper, NbS₂ is the odd one out: it does not have a CDW at any temperature.

I found the manuscript a pleasure to read. The experimental results are laid out clearly and comprehensively, covering a relevant and diverse set of experimental conditions. Transition metal-dichalcogenides are an active topic of current experimental and theoretical research and this work is a very worthy contribution to the field, both in terms of I believe it should be published in Nature Communications. I do have some questions and remarks, which follow below.

We thank the reviewer for careful reading of our manuscript and constructive comments, enabling us to improve our work. We have taken the reviewer’s comments seriously, and have revised our manuscript, accordingly. Specific points are addressed below. We hope with these changes, the reviewer will recommend our manuscript for publication.

* Around line 70: This paragraph contains a discussion of mechanisms contributing for or against CDW. However, the reason why this class of materials wants to form a CDW in the first place is left rather open here.

We thank the reviewer for raising this critical issue. We believe that the charge density wave (CDW) in transition metal dichalcogenides (TMDs) stems from two facts. The first is the quasi-two-dimensionality of electronic properties in these systems, which is itself due to the layered nature of their crystal structures. The weak van der Waals bonding between the constituent layers allows each unit to keep its characteristics to a large extent and at the same time let the phonon vibrations to effectively rattle such weakly connected sheets. Secondly, as we have discussed in our paper, metallic TMDs, including *2H-MX*₂ compounds

studied here, suffer from a deficient charge transfer from the transition metal cations to the chalcogen anions. To compensate for this deficiency, the chalcogen p orbitals try to form additional covalent bonding with each other that when combined with existing phonon vibrations, can propagate a charge density modulation throughout the whole system. This combination, in our opinion, is the key factor in the emergence of CDW in metallic TMDs.

To clarify this, we have added the following explanation in the introduction part of our revised manuscript on page 3:

Two constraints facilitate the emergence of CDW instability and its diversity in this group of materials. The first is the fact that TMDs, in general, possess a quasi-2D electronic structure regardless of their thickness. This is due to the weak van der Waals stacking of MX_2 layers, allowing resonance of phonon vibrations along certain crystalline directions. Second, the metallic TMDs, as will be discussed later, exhibit a strong tendency to form additional covalent bonding between their chalcogens to compensate for the deficient charge transfer from the existing transition metals. When combined, these two can modulate the otherwise uniform charge distribution of metallic bands, such that the distorted state is chemically more robust.

* Around line 205: Is this analysis based on the equation $\lambda = 2g^2/\omega_{\text{phonon}}$? In that case, it might be useful to include this formula.

* More generally, including a clearer definition of the electron-phonon coupling λ in the main text could be helpful, also to put Fig 4b into context.

That is correct. In principle, this is the equation used for treating electron-phonon coupling λ . More precisely, we have used the implementation developed by Poncé *et al.* (Reference 50 in the revised manuscript) for this purpose. To clarify this, We have added a detailed description in the ‘‘Methods’’ part of our revised manuscript to elaborate on the methodology used for the calculation of λ . It reads:

In this scheme⁵⁰, λ is obtained by summing up the electron-phonon coupling arising from each individual phonon mode ν at all available phonon wave vectors \mathbf{q} , i.e. $\lambda = \sum_{\mathbf{q}\nu} \lambda_{\mathbf{q}\nu}$. $\lambda_{\mathbf{q}\nu}$ is expressed as follows,

$$\lambda_{\mathbf{q}\nu} = \frac{1}{N(\varepsilon_F)\omega_{\mathbf{q}\nu}} \sum_{nm} \int_{BZ} \frac{d\mathbf{k}}{\Omega_{BZ}} |g_{mn}^\nu(\mathbf{k}, \mathbf{q})|^2 \delta(\varepsilon_{n\mathbf{k}} - \varepsilon_F) \delta(\varepsilon_{m\mathbf{k}+\mathbf{q}} - \varepsilon_F)$$

where $\omega_{\mathbf{q}\nu}$ is the phonon frequency for mode ν at the wave vector \mathbf{q} ; $\varepsilon_{n\mathbf{k}}$ is the eigenvalue of the n -th Kohn-sham eigenfunction $\psi_{n\mathbf{k}}$ at wave vector \mathbf{k} in the BZ; ε_F is the Fermi level; $N(\varepsilon_F)$ corresponds to the density of states per spin at $N(\varepsilon_F)$ and Ω_{BZ} is the volume of the BZ. Above, $g_{mn}^\nu(\mathbf{k}, \mathbf{q})$ is the first-order electron-phonon matrix element defined as,

$$g_{mn}^\nu(\mathbf{k}, \mathbf{q}) = \frac{1}{\sqrt{2\omega_{\mathbf{q}\nu}}} \langle \psi_{m\mathbf{k}+\mathbf{q}} | \partial_{\mathbf{q}\nu} V | \psi_{n\mathbf{k}} \rangle$$

with $\partial_{\mathbf{q}\nu} V$ being the first derivative of the self-consistent potential associated with phonon $\omega_{\mathbf{q}\nu}$.

The cited reference is,

⁵⁰ Poncé, S., Margine, E. R., Verdi, C. & Giustino, F. *EPW: Electron–phonon coupling, transport and superconducting properties using maximally localized Wannier functions. Comput. Phys. Commun.* **209**, 116–133 (2016).

* The authors have performed ab-initio calculations, for the band structure and the electron-phonon coupling. I would be interested to know about the ab-initio phonon spectra and their relation with the experimentally observed Raman spectra.

Following the reviewer’s suggestion, we have included the phonon dispersions for all four monolayer compounds in a new section (Section 2) in our Supplementary information. To connect them with our experimental data, we have added legends in each panel to indicate the corresponding A_{1g} and E_{2g} branches, appearing in our Raman spectroscopy measurements as active phonon modes. As can be seen the agreement with experiment is satisfactorily good, and many key features such as reversal of phonon modes in NbSe₂ is well reproduced.

The changes in the main text is as follows: In the first paragraph on page 5 after “As shown in Figure 2c, both the A_{1g} and E_{2g} modes are found to have much higher frequencies in S-based $2H-MX_2$ compounds than in Se-based ones.” we have added *Similar results are found for the monolayer samples, also consistent with our first-principles calculations (Supplementary Section 2).*

Also, below is the new Supplementary Section 2:

2. Comparison of phonon dispersions

Figure S3 summarizes the calculated phonon dispersions for all four monolayer $1H-MX_2$ ($M=Nb, Ta$ and $X=S, Se$) compounds. In a glance, one can notice a systematic difference between the S-based and Se-based compounds. In the former, the optical modes appear at much higher frequencies. Also, the gap between the A_{1g} and E_{2g} modes is much larger in the S-based compounds than in the Se-based ones. As described in the main text, such a hardening of phonon modes is an indication that the chemical bonding in S-based compounds is more ionic (and thus stronger) than that in Se-based compounds. Overall, the calculated values for A_{1g} and E_{2g} modes at Γ point (i.e. $\mathbf{q} = (0, 0, 0)$) agree reasonably well with our experimental data, as shown in Figure S4. The largest deviation is found for NbS₂ where the calculations underestimate the E_{2g} mode by $\sim 19\%$. We attribute this to the localized nature of wave-functions in this material, requiring exchange-correlation approximations beyond standard density functional methods such as Perdew-Burke-Ernzerhof functional used in this study as well as more accurate pseudo-potentials. Nevertheless, our calculations provide a strong confirmation for many key observations in this study, including the fact that NbSe₂ is the only compound in this family in which A_{1g} mode lies below the E_{2g} mode (see for example Figure 2 in the main text). ”

Figure S3: *Calculated phonon dispersions for monolayer 1H-MX₂ (M=Nb, Ta and X=S, Se). The red squares and green circles indicate the expected locations of the experimentally observable A_{1g} and E_{2g} modes, respectively.*

Figure S4: *Raman-active phonon modes for monolayer 1H-MX₂ (M=Nb, Ta and X=S, Se). (a) Comparison of the measured A_{1g} mode frequency analyzed from the data at 4 K in Figure S5 and the calculated mode frequency taken from Figure S3. (b) The corresponding results for the E_{2g} mode.*

* In the discussion of Ref 34, around line 255: Hall et al. (ACS Nano 2019, 13, 9, 10210–10220) discuss the role of the environment in TaS₂ monolayers in more detail. They find that it is not just charge transfer but also hybridization between monolayer and substrate that is responsible for the absence of CDWs on gold substrates.

We thank the reviewer for pointing out this important reference. We have included it and two more related works on 2H-TaS₂, and expanded the corresponding discussions in the revised version. It now reads:

Interestingly, CDWs were found to be absent in epitaxial monolayer TaS₂ on Au(111) substrate³⁷, but persist in MBE-grown monolayer TaS₂ on graphene/Ir(111)³⁸ as well as in the exfoliated monolayer studied here. These findings provide further evidence that the CDW formation in such atomically thin TMDs is, in general, highly susceptible to the surrounding environment, which could be a neighboring layer or even a substrate. As we discussed earlier, the deficient charge transfer in metallic TMDs makes them active in finding new pathways for electron hopping so that the chalcogen *p* orbitals can gain all electrons they need to form a closed shell system. Such pathways could, in principle, extend to a substrate. As an example, with Au(111) surface as a substrate, one can expect electron charge transfer from spatially extended 4*s* electrons of Au to the chalcogens above them. Meanwhile, the trigonal symmetry of Au(111) substrate enforces the original symmetry of pristine monolayer TaS₂, thereby avoiding any lattice deformation. Of course, hybridization between monolayer TaS₂ and the Au(111) substrate could significantly affect its electronic band structure^{38–40}, and correspondingly, its intrinsic properties. Alternatively, when the substrate is made of chemically inactive orbitals (such as *p*_π states in graphene), the CDW is still inevitable.

The cited references are as follows,

- ³⁷ Sanders, C. E. et al. Crystalline and electronic structure of single-layer TaS₂. *Phys. Rev. B* **94**,335081404 (2016).
- ³⁸ Hall, J. et al. Environmental control of charge density wave order in monolayer 2H-TaS₂. *ACS Nano* **13**, 10210–10220 (2019).
- ³⁹ Shao, B. et al. Pseudodoping of a metallic two-dimensional material by the supporting substrate. *Nat. Commun.* **10**, 180 (2019).
- ⁴⁰ Lefcochilos-Fogelquist, H. M., Albertini, O. R. & Liu, A. Y. Substrate-induced suppression of charge density wave phase in monolayer 1H-TaS₂ on Au(111). *Phys. Rev. B* **99**, 174113 (2019).

Inspired by the reviewer’s comments and the works cited above, we think it will be interesting to further investigate the effects of the substrate on the CDW order in this family of materials. The data on the suspended thin flakes, shown in the original Supplementary Section 10, will form the basis of such a study. We are afraid that they will go largely unnoticed if they only appear in a supplementary file. If the reviewer thinks it is acceptable, we would like to remove these data from the Supplementary Information and leave them for a future publication. We believe this will serve as a better contribution to the field. Accordingly, we may change the following sentence in the Discussion section, “The exfoliated samples are expected to exhibit intrinsic CDW properties, verified by control experiments on suspended thin flakes (Supplementary Section 10).”, with the content in the parenthesis replaced by “data to appear in a future publication”.

* It would be useful to compare these Raman spectroscopy results with those available in the recent literature, e.g. *Phys. Rev. Lett.* 122, 127001(2019), *Phys. Rev. B* 100, 165414

(2019), Phys. Rev. B 98, 165109 (2018) and Phys. Rev. B 97, 094502 (2018).

We appreciate the reviewer for suggesting to cite these works. They indeed help our discussion. Phys. Rev. Lett. 122, 127001 (2019) and Phys. Rev. B 97, 094502 (2018) indeed support our assignment of the CDW amplitude modes and zone-folded modes in 2H-TaS₂ and 2H-NbSe₂, respectively. We add them as the new Refs. 30 and 31 in the revised version and cite them in the original Line 118.

Phys. Rev. B 100, 165414 (2019) is on the study of 1T-TaS₂, which found a suppression of both the commensurate and the incommensurate CDW transitions in atomically thin sample. This is consistent with the results from an electrical transport study on the same system, as shown in the original Ref. 18. We therefore cite these two works together in the introduction.

Phys. Rev. B 98, 165109 (2018) mainly focuses on the layer-number and excitation energy dependence of the first-order phonons in NbSe₂. The established layer-number dependence of the A_{1g} and E_{2g} phonon frequencies are consistent with ours shown in the Supplementary Information. We therefore cite this work for comparison.

Changes that we have made are as follows:

(1) The original Line 118 is modified to *These are the amplitude modes or zone-folded modes unique to the CDW phase*^{8,28–31}. adding two more references,

³⁰ *Grasset, R. et al. Higgs-mode radiance and charge-density-wave order in 2H-NbSe₂. Phys. Rev. B 97, 094502 (2018).*

³¹ *Grasset, R. et al. Pressure-induced collapse of the charge density wave and Higgs mode visibility in 2H-TaS₂. Phys. Rev. Lett. 122, 127001 (2019).*

(2) In the original Lines 44–47, “Mechanical exfoliation^{8,17,18}, molecular-beam epitaxy (MBE)^{7,19–21}, and chemical vapour deposition^{22,23} have produced a plethora of 2D materials exhibiting CDWs. Dimensionality reduction was shown to enhance the CDW order in some of them^{8,19,21,23} while suppress it in others^{7,18,22,24}.” we add Phys. Rev. B 100, 165414 (2019) as the new Ref. 19.

¹⁹ *Ramos, S. L. L. M. et al. Suppression of the commensurate charge density wave phase in ultrathin 1T-TaS₂ evidenced by Raman hyperspectral analysis. Phys. Rev. B 100, 165414 (2019).*

(3) In the original Supplementary Section 3.1, when discussing the layer-number dependence of the phonon mode frequency, we change “This has been well established in other similar layered materials such as 2H-MoS₂².” to *This has been well established for NbSe₂² as well as in other similar layered materials such as 2H-MoS₂³.*” adding the new Ref. 2,

² *Hill, H. M. et al. Comprehensive optical characterization of atomically thin NbSe₂. Phys. Rev. B 98, 165109 (2018).*

*Smaller comments:

* Fig 1 and page 3: “planner hopping” → “planar hopping”?

In the revised manuscript, we have corrected these typos.

* Fig 1 (g): I would write the label as “bulk has harder/softer lattice”, so that the panel can be understood without the caption. In the current version, it is not clear which situation has the harder lattice.

If the reviewer does not mind, we wish to keep them unchanged. The positive (negative) ΔQ_σ region corresponds to a harder (softer) lattice as compared to a monolayer. This terminology is thus not limited to the bulk. It is applicable, for example, to the bilayer too, and in general any configuration beyond monolayer. That is why we simply write “harder lattice” and “softer lattice” and highlight each regime by different colours in the figure.

* Fig 1f and similar figures: It does not really make sense to connect the data points by lines here, and I find it a bit distracting.

We totally agree with the reviewer. Those line do not bear any physical meaning. In the revised manuscript, we have replaced the lines and dots in Figure 1f and 1g, Figure 2c and Figure 4b and represented our results with bar charts.

* Line 92, “Pierels” → “Peierls”

Corrected in the revised manuscript.

RESPONSE TO REVIEWER 2

This work experimentally investigates the CDW transition in thin exfoliated sheets of NbSe₂, TaSe₂ and TaS₂, based upon the emergence of the corresponding amplitude modes in the Raman spectrum. A major finding is that T_{CDW} increasing (decreases) with decreasing sheet thickness for NbSe₂ (TaSe₂ and TaS₂).

As there are indeed partially contradictory theoretical predictions and also experimental observations reported in the literature on CDWs in this (and similar) type(s) of material, the present study makes a novel and valuable contribution to this field. The detailed Raman experiments seem to be carried out carefully (it is especially important that the bulk crystals were exfoliated in a glove box and subsequently encapsulated by h-BN), and also the data evaluation looks reasonable.

Furthermore, the three parameter-based model (fig. 1a) proposed to account for the presence or absence (like in NbS₂) of CDW instability, and the thickness dependence of T_{CDW} for the three different compounds, makes sense and is — despite its simplicity — in quite good agreement with the experimental observations. Although no fundamentally new theoretical result is provided, the presented model can nevertheless help to further clarify relevant issues on CDWs in 2D materials. Finally, also the mentioned experimental factors (in particular the role of the substrate) are indeed likely to have an influence on T_{CDW} in these compounds.

We appreciate the reviewer's time to evaluate our work and are grateful for the positive comments regarding the novelty and significance of our results.

REVIEWERS' COMMENTS:

Reviewer #1 (Remarks to the Author):

I am satisfied with the changes and believe the manuscript is ready for publication.

REVIEWERS' COMMENTS:

Reviewer #1 (Remarks to the Author):

I am satisfied with the changes and believe the manuscript is ready for publication.

We thank the reviewer again for evaluating our work.